# Seasonal Occurrence of the Indian Ocean Blue Whale (*Balaenoptera musculus indica*) off South Coast of Sri Lanka

Upul S. P. K. Liyanage [1,2,*], Pradeep K. P. B. Terney [1], Upali S. Amarasinghe [3], Kanapathipillai Arulananthan [4] and Marianne Helene Rasmussen [5]

1   Department of Oceanography and Marine Geology, Faculty of Fisheries andMarine Sciences & Technology, University of Ruhuna, Matara 81000, Sri Lanka; terney@fish.ruh.ac.lk
2   National Aquatic Resources Research and Development Agency, Regional Research Center, Kapparatota, Weligama 81700, Sri Lanka
3   Department of Zoology and Environmental Management, Faculty of Science, University of Kelaniya, Kelaniya 11600, Sri Lanka; zoousa@kln.ac.lk
4   National Aquatic Resources Research and Development Agency, Crow Island, Mattakuliya, Colombo 01500, Sri Lanka; k.arulan@gmail.com
5   Húsavík Research Centre, University of Iceland, 640 Húsavík, Iceland; mhr@hi.is
*   Correspondence: upulliyanage@hotmail.com; Tel.: +94-718360025

**Abstract:** This paper describes the distribution, abundance and seasonal variation in blue whales (BWs) on the south coast of Sri Lanka, off Mirissa, where they are faced with anthropogenic threats. Data collection encompassed opportunistic sightings by whale-watching (WW) operations. This study revealed that large aggregations of BWs consistently overlapped with busy shipping lanes located between Dondra Head and Galle within important foraging and breeding regions on the south coast. Throughout 2015 (except June and July), 729 BWs were sighted over 177 sighting days. The sighting frequency was higher during the northeast monsoon and the first intermonsoon and lower throughout the second intermonsoon (*n* = 9) and latter part of the southwest monsoon. The highest frequencies of BW encounters per day occurred in April (*n* = 15) and December (*n* = 20), while the mean annual group size per sighting was 3.07 ± 0.24. From the January-to-April season, 13 mother–calf combinations and 1 pregnant cow were sighted off the southern coastline of Mirissa, suggesting the calving season peaks between the months of March and April in Sri Lanka. As this important habitat overlapped with the busiest shipping lanes, fishing and commercial whale-watching activities, the authorities have to take action toward the conservation of this ecosystem and whales as well as their safe navigation.

**Keywords:** blue whales; *Balaenoptera musculus indica*; distribution; abundance; Sea Surface Temperature (SST); Sri Lanka

## 1. Introduction

Previous studies have shown that Sri Lanka is an important cetacean habitat, with high species richness and encounter rates, especially of blue whales (BWs) (*Balaenoptera musculus indica*) [1–4]. Moreover, Sri Lanka is located at the centre of the Indian Ocean Marine Mammal Sanctuary (IOMMS) between the latitudes 5°55′ and 9°51′ N and longitudes 79°41′ and 81°53′ E, south of the Indian subcontinent, separated by the narrow Palk Strait. Oceanic processes around the island are driven by two major monsoonal winds in combination with bathymetric features (i.e., submarine canyons), all of which produce intense upwelling [5], which, in turn, provide ideal foraging conditions for aggregations of BWs [6]. An increasing number of previous studies, e.g., [7–10], have shown that BWs are a group of stenophagous predators and exclusively prefer to make feeding aggregation on productive upwelling regions, where patchy and dense krill aggregations occur. Gill et al. [11] revealed that the sea surface temperature and other various seascape variables associated with upwelling

played a significant role in the Bonney upwelling region used by blue whales for feeding. Moreover, frequent aggregations of BWs along the edge of the Monterey Bay submarine canyon were also documented [7].

BWs are a highly migratory group of cetaceans, and their migration has been considered long-distance travel on an annual or seasonal basis, following local climatic changes, prey abundance or for breeding purposes [12,13]. Even though many BW populations follow long-range migration patterns, those inhabiting Sri Lankan waters are recorded throughout the year [4,14]. However, spatial and temporal variations might be expected regarding prey availability. Mirissa on the south coast, Mullaithivu and Trincomalee on the east coast and Kalpitiya on the northwest coast have been identified as three major BW hotspots in Sri Lanka [15], although BWs on the coast of Mirissa are faced with multiple threats in comparison with the other sites [16,17]. Within the past two decades, an amalgamation of anthropogenic threats, such as increased vessel traffic, ship strikes, acoustic pollution, maritime activities, and commercial WW operations, have significantly increased in the area [4,17–19]. It has been estimated that annually around 40,000–50,000 ships running in an east–west direction pass this area.

Further, the coast of the Mirissa area has been identified as one of the best BW watching sites in the Northern Indian Ocean region. The marine sector of global tourism is considered a blooming industry, which is predicted to become one of the value-adding segments of the ocean economy by 2030 [20]. Among the different components of marine ecotourism, whale and dolphin watching is playing a vital role, giving substantial economic benefit to the 119 coastal nations in the world while providing viewing opportunities for local and foreign tourists [21–23]. Further, it is considered a way to build awareness of the conservation of the threatened species while extracting noncommercial values. Sri Lanka is a developing country, and tourism development has rapidly influenced the expansion of WW operations, and the 'socio-economic value' of BW populations has steadily increased [16,24]. Although the whale-watching industry in Sri Lanka may provide future conservation opportunities and socioeconomic benefits, the policymakers remain apprehensive that WW activities may deleteriously impact BW populations [17,21,25]. Furthermore, the adoption of unsustainable whale-watching practices due to the extensive competition between boat operators during peak seasons is often the result of economic pressures to achieve short-term benefits from ecotourism at the expense of displacing free-ranging BWs into busy shipping lanes and thus increasing the risk of ship collisions [16,17,24].

The implementation of a conservation and management regime for marine predators and their habitats and an understanding of the oceanographic processes that drive prey abundance are critically important [26]. Moreover, information on the movement, nature of the habitat occupancy of the species and the spatial and temporal scales variations in distribution are also critical factors to be considered. Contemporary data pertaining to encounter rates, distribution and abundance are, therefore, urgently required to evaluate the impacts of anthropogenic threats on data-deficient BW populations within industry-dominated waters [27]. Long-term studies on the abundance, distribution and seasonal migration patterns of cetacean are very important for the management and conservation of threatened marine mammal species. In the present study, an attempt is made to investigate the abundance, distribution and seasonal occurrence of BWs off Mirissa in southern Sri Lanka. Due to the lack of year-round dedicated BW expeditions off the southern coast of Sri Lanka, the data drawn from the WW tour operator's log book were used for this study.

## 2. Materials and Methods

Over the last three decades, Sri Lankan waters have been identified as favorable foraging and the breeding grounds for many species of cetacean populations [28]. One of the well-known and world-renown BW 'hotspots' in Asia off coast of Mirissa in Sri Lanka was selected for studying the distribution and seasonal occurrence of BWs (Figure 1) based on their economic importance and the increased risk of anthropogenic threats [14,16,17].

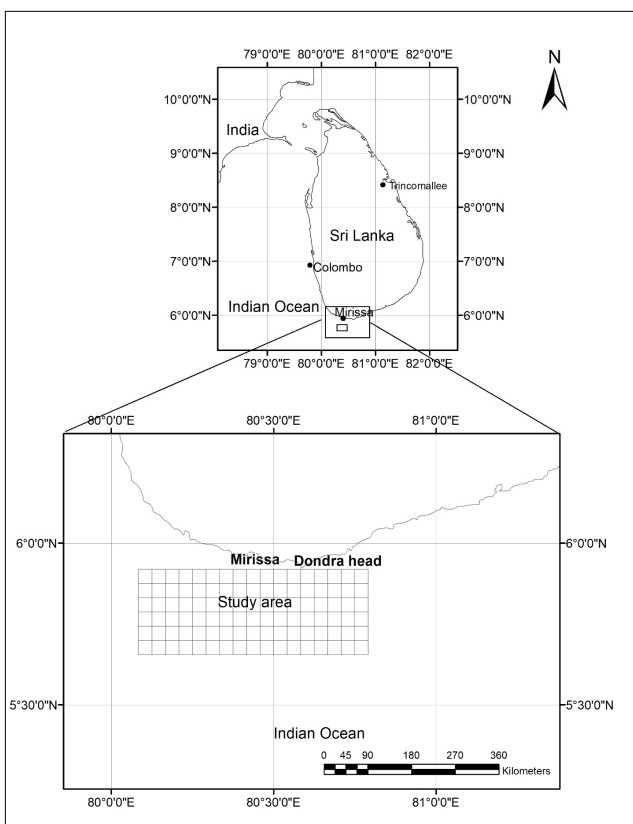

**Figure 1.** Map of the study area.

This area is on the major shipping routes and widely used for commercial WW and fisheries. Considering the high cost of dedicated cetacean surveys and logistical and technical constraints, data collated from the commercial WW operator, the 'Raja & the Whale', was used for this study. Raja & the Whale is a well-established commercial WW operator in Sri Lanka, based at the Mirissa fisheries harbor. The boat was operated every day of the year except during the peak southwest monsoon (May to July) and other days of extreme weather events. The ocean productivity and circulations pattern of the marine ecosystem of Sri Lanka are driven by reversal monsoonal winds [18] as follows:

Northeast monsoon (NE)—November to February
First intermonsoon (IM1)—March to April
Southwest monsoon (SW)—May to August
Second intermonsoon (IM2)—September to October

A commercial whale-watching boat powered by 250 Hp inboard engine and with observation flying bridge deck was used for these operations. The observation height (eye-level) was approximately four meters above the sea surface, and around 5–6 observers were able to scan all around, except the astern side. The vessel set sail at 6.30 a.m. from the Mirissa fisheries harbor and returned in 3–8 h, based on the proximity to the sighting location, currents, winds, visibility and variations in the cetacean richness; the cruising distance of a daily trip was 50–100 km. Data collection was performed between 8 and 12 a.m. on each whale-watching day, and no searching was performed after 12 o'clock. All observations were conducted with both the naked eye and confirmed with pair of Nikon (8 × 12) marine binoculars. After starting the trip from Mirissa fisheries harbor, the boat traveled 20–30 km toward the south (continental shelf) and then turned in east or west direction (along the shelf or parallel) until whales were spotted. If no whale sightings occurred, boat was returned to the harbor at around 2–3 p.m. The crew was well experienced in species identification and spotting, as they had conducted previous research work with local and international research organizations and universities. To

avoid duplicate recordings, observers carefully observed the number of whales in each pod, bearing and distance to the pod from the vessel, blow shape and interbreathing interval of some individuals, heading, diving interval, speed, behavior, body size and special marks on the body, etc. At each sighting, GPS locations, species, number of individuals (and group size), presence of mother and calf pairs and their behavior were recorded in standard record book. In addition, the boat tracking and sighting locations were also recorded using a handheld Garmin eTrax 10 GPS unit.

Sighting data were analyzed on monthly basis for studying the variations in abundance pattern. The number of individuals sighted per trip was estimated as the ratio of the number of individuals sighted per month to the number of total whale-watching trips undertaken within a particular month. Similarly, number of individuals sighted per successful trip was the ratio of number of blue whales sighted to the number of successful trips within the month.

The chlorophyll-a concentration in the water is considered a good indicator of the ocean's primary productivity, which can possibly determine the distribution and abundance of species [29,30]. Upwellings and primary productivity of the study area in the south coast were studied using the remote sensing data extracted by the National Aeronautic and Space Administration (NASA) Ocean Biology Processing Group (OBPG: http://oceancolor. gsfc.nasa.gov (accessed on 1 January 2015 to 31 December 2015)). The monthly average chlorophyll-a (CHL-a) and sea surface temperature (SST) data from January to December 2015, with a 4 km spatial resolution acquired through Aqua/Terra Modis satellites, were processed using ArcGIS (ESRI).

The impacts of SST and CHL-a on the blue whale monthly sightings were statistically tested using Mann–Whitney nonparametric analysis using Minitab (Version 19) software. For the statistical analysis, study area was divided into 102 ($6 \times 17$) blocks, with the extent of 4 km$^2$ with each block, similar to the downloaded images. Then, mean CHL-a and SST values of each block related to the SW monsoon and non-SW monsoon were extracted using the fishnet tool of the data management tool of ArcGIS. The mean SST and CHL-a values of the study area were estimated averaging the values of each block and then categorized into four groups as high and low CHL-a and SST. A value above the mean was considered high and below categorized as low. Number of sightings in each block was manually calculated, and the number of sightings during high SST and low SST, as well as during high CHL-a and low CHL-a, were compared using nonparametric Mann–Whitney test. Similarly, number of sightings during the SW monsoon and non-SW monsoon was also compared using the same statistical approach.

## 3. Results

Opportunistic sightings contiguous with the southern coastline revealed year-round variation in the abundance of BW aggregations overlapping one of the busiest shipping lanes in the world between Dondra Head and Galle. However, 'Raja & the Whale' stopped operations between 6 May and 31 July 2015 due to rough sea conditions, although other whale-watching tour operators opportunistically verified the presence of BWs throughout this survey period. In general, BWs were the most frequently encountered compared to other cetacean species in the south coastal region off Mirissa. The opportunistic blue whale sightings during the year 2015 are given in Table 1.

Of the 266 WW trips during the study period, only 177 WW days were successful. The percentile of the successful trips to the total WW trips was 68.7%. During 2015 (except a major part of May, June and July), 729 individual BWs were sighted (including resightings) over 266 WW trips, as an average 2 to 3 BW individuals were sighted per day. There is a higher likelihood of encountering BWs off the southern coast in the months of January to April and December (Table 1). The highest frequency of BW sightings ($n = 176$) occurred during the first intermonsoon season (April), and the lowest frequency of sightings ($n = 9$) occurred during the northeast monsoonal season (in November). The highest probability of BW sightings was in April (96.66%), with an average of six individuals

per day and the month of December, with an average of seven sightings per day. (A total of 104 opportunistic sightings were documented between 24 and 30 December) The month of May was not considered in the comparison due to the limited number of trips operated.

**Table 1.** Summary of blue whale sightings—opportunistic survey in 2015 on south coast (off Mirissa).

| Month | Number of Total Trips | No. of Successful Trips | No. of Sightings per Month | Successful Trips to Total Trips % | Number of Individuals Sighted | Number of Individuals Sighted per Trip | No. of Individuals Sighted per Successful Trip |
|---|---|---|---|---|---|---|---|
| Jan | 31 | 28 | 37 | 90.0 | 135 | 4.3 | 4.8 |
| Feb | 28 | 20 | 28 | 71.4 | 42 | 1.5 | 2.1 |
| Mar | 31 | 26 | 40 | 83.8 | 91 | 2.9 | 3.5 |
| Apr | 30 | 29 | 48 | 96.6 | 176 | 5.8 | 6.0 |
| May | 4 | 4 | 5 | 100.0 | 5 | 1.2 | 1.2 |
| Jun | - | - | - | - | - | - | - |
| Jul | - | - | - | - | - | - | - |
| Aug | 26 | 15 | 21 | 57.6 | 66 | 2.5 | 4.4 |
| Sep | 28 | 9 | 10 | 32.1 | 14 | 0.5 | 1.5 |
| Oct | 27 | 14 | 15 | 51.8 | 18 | 0.6 | 1.2 |
| Nov | 30 | 8 | 8 | 26.6 | 9 | 0.3 | 1.1 |
| Dec | 31 | 24 | 34 | 77.4 | 173 | 5.5 | 7.0 |
| Annual | 266 | 177 | 261 | 68.7 | 729 | 2.5 | 3.3 |

The monthly variation in the BW group sizes is given in Table 2.

**Table 2.** Monthly variation in the BW groups sizes, monthly mean, and maximum and minimum sightings.

| Group Size | 1 | 2 | 3 | 4 | 5 | 6–10 | 11–15 | 16–20 | Min. | Max. | Mean $\pm$ SE |
|---|---|---|---|---|---|---|---|---|---|---|---|
| Jan | 17 | 8 | 1 | 2 | - | 5 | 4 | - | 1 | 12 | 3.65 $\pm$ 0.67 |
| Feb | 19 | 6 | 1 | 2 | - | - | - | - | 0 | 4 | 1.50 $\pm$ 0.16 |
| Mar | 24 | 5 | 3 | 1 | 1 | 6 | - | - | 0 | 7 | 2.28 $\pm$ 0.32 |
| Apr | 14 | 9 | 3 | 9 | 5 | 6 | 2 | - | 1 | 15 | 3.66 $\pm$ 0.43 |
| May | 5 | - | - | - | - | - | - | - | 1 | 1 | 1.0 $\pm$ 0 |
| June | - | - | - | - | - | - | - | - | - | - | - |
| July | - | - | - | - | - | - | - | - | - | - | - |
| Aug | 7 | 5 | 2 | 3 | 1 | 3 | - | - | 0 | 10 | 3.87 $\pm$ 0.64 |
| Sep | 8 | 2 | - | - | - | - | - | - | 0 | 4 | 1.40 $\pm$ 0.30 |
| Oct | 12 | 3 | - | - | - | - | - | - | 0 | 2 | 1.21 $\pm$ 0.10 |
| Nov | 7 | 1 | - | - | - | - | - | - | 0 | 2 | 1.13 $\pm$ 0.12 |
| Dec | 12 | 7 | 5 | 2 | - | 3 | - | 5 | 0 | 20 | 5.09 $\pm$ 1.13 |
| Total | 126 | 48 | 18 | 23 | 12 | 23 | 6 | 5 | | | 3.07 $\pm$ 0.24 |
| % | 48.3 | 18.4 | 6.9 | 8.8 | 4.6 | 8.8 | 2.3 | 1.9 | | | |

Min. and Max. are the minimum and maximum number of BWs sighted within the month. SE is the standard error.

According to Table 2, the majority of group sizes (48.2%, *n* = 126 of 261) of BWs were observed as solitary whales. During the total survey period, the largest number of solitary BWs were encountered in March (*n* = 24) and then February (*n* = 19) or, in general, from the end of the northeast monsoon to the first intermonsoon period. Further, the heterogeneous size of groups was noticeably observed within this period more than during the other monsoon. The variation in group sizes is comparatively less during the September- to-November period. According to Table 2, August is the only month of the southwest monsoon that has a complete data set of sightings, and it reflects a high group diversity similar to the northeast monsoon. The maximum, minimum and mean group size of BW aggregations in each month tended to fluctuate drastically. In January, May and

April, at least a single blue whale was found during every WW tripy, unlike in other months (except June and July) during when not even a single blue whale was found. The highest frequency of BW encounters per day occurred in April (*n* = 15) and December (*n* = 20). The maximum average group sizes of BWs occurred in the month of December (5.09 ± SE 1.13), while the smallest group sizes occurred in the month of May (1 ± SE0), although the boat only operated for five days of this month. The highest number of monthly BW encounter rates per day occurred during the northeast monsoon (November–February) and the first intermonsoon (March–April) when compared to the southwest monsoon (May–August) and the second intermonsoon (September–October). However, a comparison with the southwest monsoon is not sensible due to the lack of data for the whole period.

The mean monthly variations in the SST and CHL-a concentrations in the study area are given in Table 3.

**Table 3.** Monthly variations in SST and CHL-a (mg/m$^3$).

| Month | SST (°C) | | CHL-a (mg/m$^3$) | |
|---|---|---|---|---|
| | **Mean ± SE** | **Range** | **Mean ± SE** | **Range** |
| January | 27.74 ± 0.04 | 27.45–28.61 | 0.410 ± 0.019 | 0.280–1.208 |
| February | 28.42 ± 0.03 | 28.01–29.04 | 0.242 ± 0.013 | 0.180–1.858 |
| March | 29.08 ± 0.02 | 28.87–29.64 | 0.140 ± 0.004 | 0.119–1.770 |
| April | 30.38 ± 0.03 | 29.85–30.98 | 0.126 ± 0.010 | 0.083–2.032 |
| May | 29.40 ± 0.05 | 28.03–29.97 | 1.107 ± 0.190 | 0.193–7.992 |
| August | 28.47 ± 0.06 | 26.56–29.02 | 0.527 ± 0.114 | 0.156–4.946 |
| September | 28.60 ± 0.05 | 27.51–29.02 | 0.334 ± 0.053 | 0.000–2.663 |
| October | 29.30 ± 0.02 | 28.83–29.66 | 0.385 ± 0.077 | 0.147–4.276 |
| November | 29.50 ± 0.03 | 29.11–30.03 | 0.238 ± 0.019 | 0.124–1.850 |
| December | 28.94 ± 0.02 | 28.73–29.44 | 0.350 ± 0.039 | 0.162–4.218 |

The monthly CHL-a and SST concentrations in the year 2015 were highly variable. According to Table 3, the highest CHL-a concentrations were recorded during the month of May 1.107 ± 0.19 (range 0.193–7.992) mg/m$^3$ and August 0.527 ± 0.1140 (range 0.156–4.946) mg/m$^3$ in the SW monsoon, and the lowest was 0.126 ± 0.01 (range 0.083–2.032) mg/m$^3$ in April and March 0.140 ± 0.004 (0.119–1.77) mg/m$^3$ in the first intermonsoon (IM1). Even though small chlorophyll concentrations were recorded during the March and April BW sightings, they were comparatively high (Figure 2). According to Figure 2, the high CHL-a concentrations observed close to the coastal line might be the coastal upwelling induced by the river inflow (Polwathumodara and Nilwala) to Weligama Bay and Matara. In contrast, this high sea surface CHL-a distribution was limited to the area from the 1000 m isobath to the coastline.

Spatial variations in the SST during the year 2015 are shown in Figure 3.

Figure 3 reveals high SST values close to the coastal line during the nonsouthwest monsoon period, although during the SW monsoon, the SST is lower than the deep sea.

The statistical analysis (Mann–Whitney test) performed revealed that significantly higher sightings occurred within the blocks of low CHL-a levels (U = 277.5; *p* = 0.005), and within the blocks with a high SST (U = 114.5; *p* = 0.001). However, there was no significant difference between the number of sightings within the SW monsoon and the non-SW monsoon (Mann–Whitney test; U = 311.0; *p* = 0.219).

The annual relative abundance and distribution of the BWs around the traffic separation scheme (TSS) below the south end of Sri Lanka are shown in Figure 4.

The most prevalent abundance of BWs contained a mean group size of >five members, and they had a tendency to congregate at approximately 1000 m isobaths near the continental shelf (Figure 4), and dense aggregations were observed in between the traffic separation scheme. The southern tip of Sri Lanka between Dondra Head and Galle where BWs are aggregating is known as part of one of the busiest commercial shipping lanes in the world [4].

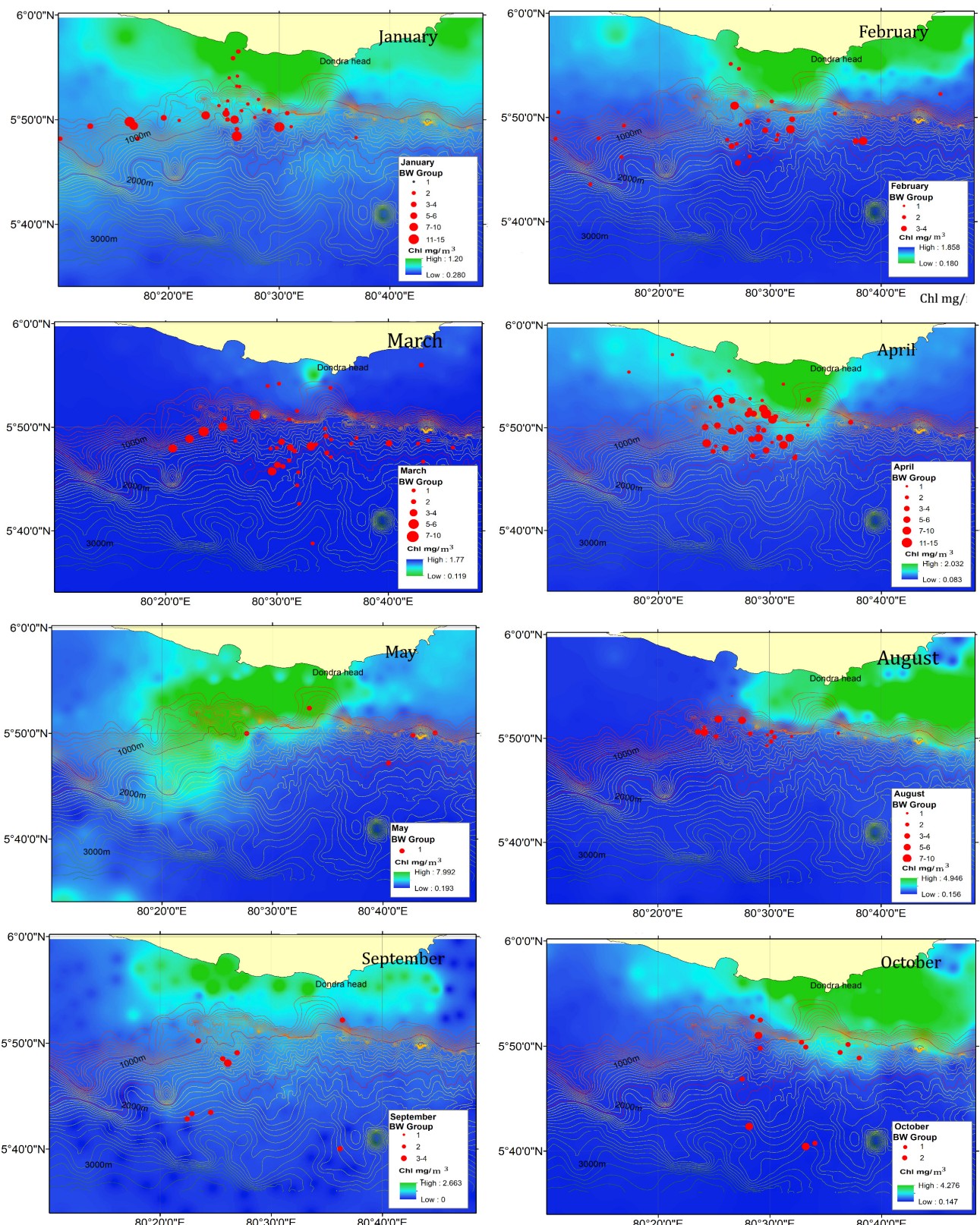

**Figure 2.** Monthly spatial variation in chlorophyll distribution off south coast of Sri Lanka.

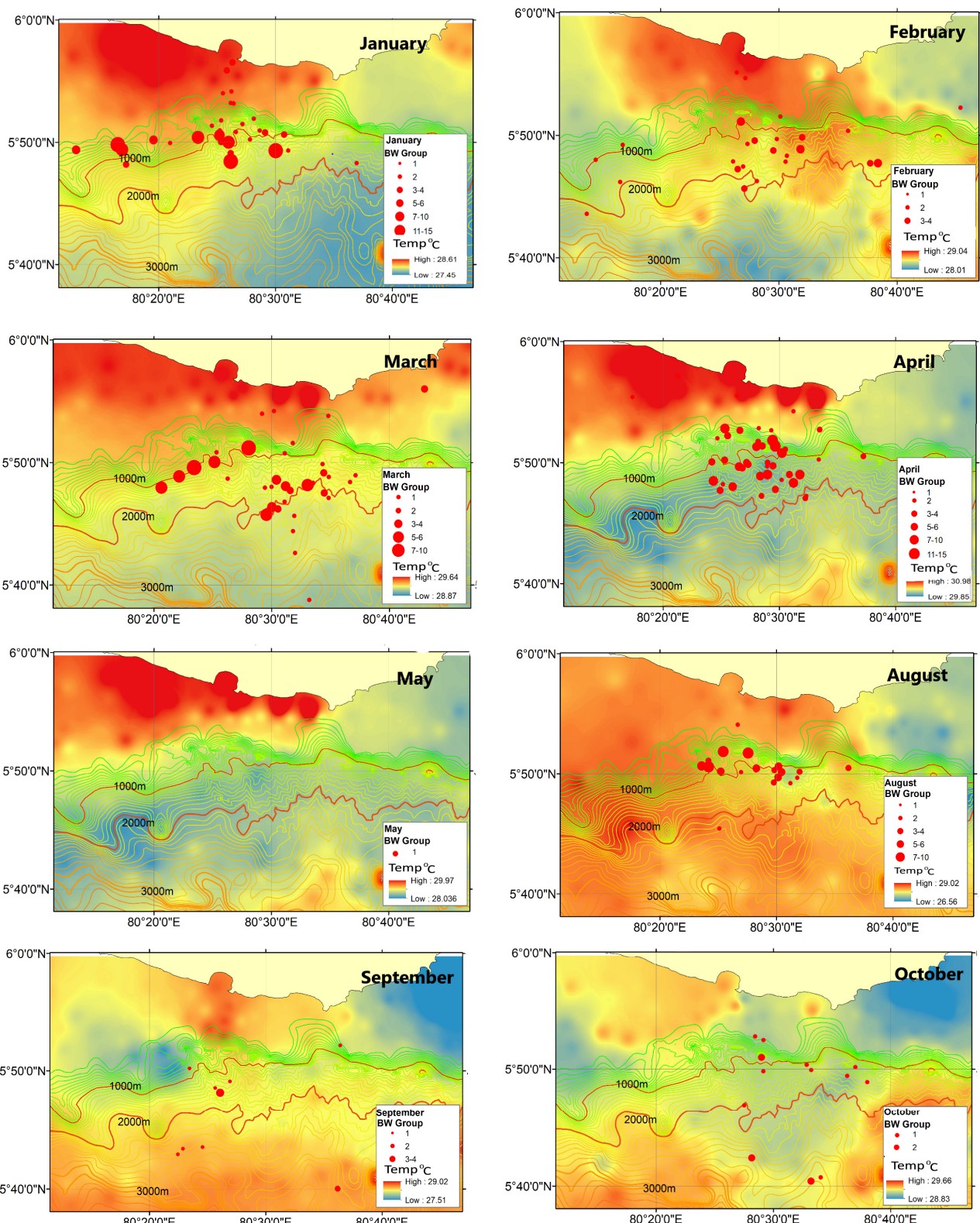

**Figure 3.** Monthly variation in SST distribution off south coast of Sri Lanka.

*Occurrence of Pregnant Cows and Mother Calf Combinations*

During the commercial WW operations in 2015, 13 sightings of mother–calf pairs and 1 pregnant female were documented between January and April in 2015 within the waters off Mirissa (Table 4).

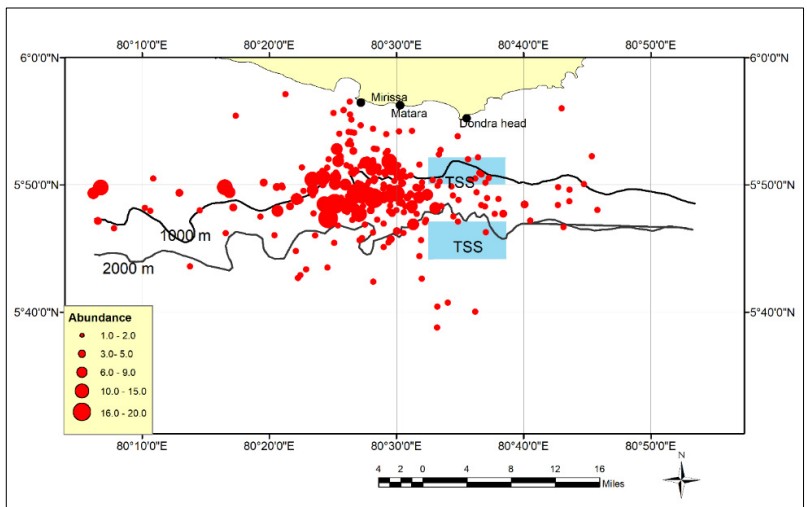

**Figure 4.** Distributional patterns and relative abundance of BW encounters off Mirissa, in the southern province, Sri Lanka. TSS is the traffic separation scheme.

**Table 4.** Records of mother–calf combinations and pregnant cows off Mirissa in 2015. NE—northeast monsoon; IM1—first intermonsoon.

| No. | Date | Monsoon | Latitude | Longitude | Remarks |
|---|---|---|---|---|---|
| 1 | 2 January 2015 | NE | 05°51.630′ N | 80°30.738′ E | Pregnant cow |
| 2 | 3 March 2015 | IM1 | 05°50.821′ N | 80°25.325′ E | |
| 3 | 4 March 2015 | IM1 | 05°45.634′ N | 80°31.936′ E | |
| 4 | 5 March 2015 | IM1 | 05°51.630′ N | 80°43.596′ E | |
| 5 | 7 March 2015 | IM1 | 05°54.199′ N | 80°30.207′ E | |
| 6 | 16 March 2015 | IM1 | 05°38.761′ N | 80°33.201′ E | |
| 7 | 22 March 2015 | IM1 | 05°48.808′ N | 80°34.871′ E | |
| 8 | 27 March 2015 | IM1 | 05°51.185′ N | 80°28.062′ E | |
| 9 | 28 March 2015 | IM1 | 05°50.045′ N | 80°25.170′ E | Same whales sighted in previous day |
| 10 | 13 April 2015 | IM1 | 05°57.119′ N | 80°21.262′ E | |
| 11 | 14 April 2015 | IM1 | 05°47.250′ N | 80°28.480′ E | Same whales sighted on previous day |
| 12 | 15 April 2015 | IM1 | 05°47.053′ N | 80°32.242′ E | |
| 13 | 18 April 2015 | IM1 | 05°47.781′ N | 80°29.671′ E | |
| 14 | 21 April 2015 | IM1 | 05°52.214′ N | 80°25.262′ E | |

Two of these fourteen encounters (whales sighted on 28 March and 14 April) were identified as the same mother and calf seen on previous days, and they did not stay in the same area for a long time.

Sighting locations of the mother calf combinations and pregnant cow are shown in Figure 5.

Seven of the thirteen records of mother–calf pairs (53%) were encountered in waters below 1000 m, and only one occurred in a deeper location far away from the land. Breeching was frequently observed in the months of November to December. At the end of December, a group of around 20 blue whales were observed for five consecutive days, showing signs of mating behaviors.

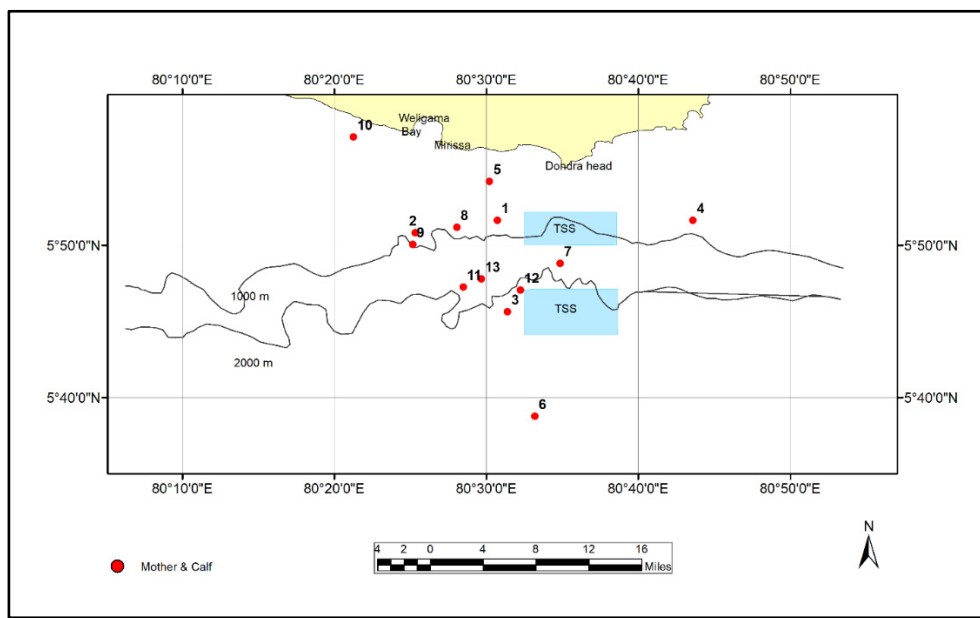

**Figure 5.** Sighting of pregnant cow (1) and mother–calf pairs (2–13) off Mirissa, south coast of Sri Lanka. TSS is the traffic separation scheme.

## 4. Discussion

BWs were frequently recorded off Mirissa in Sri Lanka throughout the year, and high frequencies were recorded during the months of December, January and April. Moreover, sightings were distributed mainly along the continental shelf margin from Dondra Head to Galle. Previous studies also revealed that the highest frequency of BW sightings in Sri Lanka transpired between Dondra Head and Galle [4,14,15] and off Mullaitivu to Batticaloa [3] on the east coast. The highly productive upwelling of nutrients and high plankton bloom productivity are both influenced by the strong southwest monsoon and the bathymetric features of Dondra Canyon, which provide a critical foraging habitat for dense aggregations of BWs [31,32]. According to Anderson et al. [33], the largest aggregations of BWs in Sri Lanka tended to occur during periods of high productivity between June and September but significantly decreased during periods of low productivity throughout the northeast monsoonal season. Meanwhile, in a cetacean survey in the Maldivian waters in the 2003–2004 northeast monsoon season, no BWs were recorded [33]. The Fridtjof Nansen ecosystem survey in the 2018 SW monsoon revealed large BW aggregations on the south and west coast of Sri Lanka [15]. However, the opportunistic sighting data for this study in 2015 revealed that the highest probability of monthly sightings and the mean number of sightings per day were highest during the northeast monsoonal and first intermonsoonal seasons (November to April), although this comparison may not be comprehensive, as the major part of the southwest monsoon (early March to the end of July) was not covered by the survey. Randage et al. [14] studied the opportunistic sighting data of the WW logbooks of the same company for the 2010–2012 period, and the results are consistent with the current study. These findings also partly fit with the results of acoustic recordings in Sri Lanka [34], but the data were missing from May to July. Hence, it was not possible to compare the entire period of the southwest monsoon. The pygmy BWs with Sri Lankan call types reached their peak in April to June but are rare during the latter part of the southwest and second intermonsoon periods [34–36]. Offshore aggregations of BWs are also fairly common and localized within the Sri Lankan, Maldivian, and Indian waters [37]. Russian whalers also encountered BWs in the Maldives throughout November; however, the majority of BW encounters in Sri Lanka, India and the Maldives typically occurred in the month of June [37,38]. In general, the BW encounter rate is a rarity throughout the month of June in the Maldives but frequently occurs throughout the months of November to April [38], whereas BW encounters occur most frequently in the northeast

of Trincomalee during the months of December to April [1–3]. The pattern of encounter rates, therefore, could indicate that a subdivision of the Sri Lankan BW population forages in the Arabian Sea during the southwest monsoon and thereafter migrates to the Maldivian waters throughout the month of April [32]. However, line transect surveys conducted in the Maldives (April, 1998) revealed four individual BW encounters over 20 days (0.2 BWs per day) and recorded smaller sighting rates than this study [38]. Russell et al. [39] recorded 21 individual BWs within the 18-day (1.16 BWs per day) survey period in the south coast region during the southwest monsoon in 2017, but the mean group size did not deviated much compared with the current study. A study conducted by 'Raja & the Whale' on the distributional patterns of BWs inside commercial shipping lanes revealed that from the total of 485 BW sightings, the average sightings per day was approximately 4.56, which was slightly higher than the average sighting per day of 4.47 found in the current study [14].

According to the current study, a large number of sightings were recorded in the blocks where medium CHL-a values were recorded along the continental shelf. The high CHL-a regions with shallow depth areas (below 300 m) were not considered for the analysis because BWs are not usually sighted in shallow waters close to the coast. The BW habitat of the south coastal region has been considered one of the upwelling areas in Sri Lanka, which is induced by the southwest monsoonal winds and coastal currents. Furthermore, the sharp and steep continental shelf margin with submarine canyons also influences the year-round primary productivity of the area [31]. Biological productivity in an upwelling ecosystem plays a vital role in energy transferring from plankton to larger vertebrate species like fish, sea birds and marine mammals [40,41]. In general, the distribution and abundance of a species depends on the distribution of its prey and in marine ecosystems, the oceanographic variables, especially the SST and CHL-a, which act as a proxy for primary production, prey abundance and their distribution [42]. Elevated CHL-a concentrations usually coincide with phytoplankton bloom [43] and are considered the first link in the trophic chain [42]. BWs exclusively feed on small crustaceans known as krill [44–46], the distribution of which is determined by the oceanographic processes and bathymetry of the area [13,47–49]. There appears to be a certain time-lag between CHL-a concentration and krill abundance, as the latter consumes algal blooms, leading to a considerable reduction in the CHL-a concentration in the area at the time the whales are present [50]. As the CHL-a concentration along the water column is determined by the interaction of the local oceanography and bathymetry of the area, the surface CHL-a value might not be representing the true concentration. For this study, we used the surface CHL-a and SST levels due to limitations in the sample collection, although krill density and BW abundance occur not only on the surface, but CHL-a levels are also in the water column [51,52]. Bedriñana-Romano et al. [53] revealed that CHL-a concentrations do not necessarily correlate with krill availability because krill densities are determined by advection with currents, vertical movements, food availability, the avoidance of predators or the grazing impacts of the plankton.

Previous studies suggest dense aggregations of BW populations encountered within lower sub-Antarctic latitudes tend to forage along the contour lines of submarine canyons, which are highly productive upwelling regions [8,44]. The migratory patterns of BWs most likely reflect the seasonal patterns of high productivity within foraging hotspots [11,30]. The migration patterns of BWs in Sri Lanka have not been well studied. Future research must be extended toward the utilization of satellite tagging to acquire a better understanding of BW migration patterns. Ever since the 1980s, opportunistic sighting data in Sri Lanka suggest the presence of a year-round resident population of BWs inhabiting the IOMMS [8,32,37]. The year-round occurrence of BWs in Sri Lanka is well documented, although breeding and calving information is rare.

During the current study period, 13 incidents of mother–calf pairs and a single pregnant cow were sighted off Mirissa in Sri Lanka. The majority of these sightings (*n* = 13) was limited to the IM1 (March–April) period. The historical published information on mother–calf sightings is rare, although Deraniyagala [54] encountered a solitary female giving birth

to her offspring in Trincomalee harbor in 1938. Furthermore, two other mother–calf pairs were also documented in the region off Trincomalee (February 1983 and March 1984) [3]. During the geophysical survey conducted off the south coast of Sri Lanka from July to August 2017, 37 BWs were recorded, although no mother–calf pairs were observed [40]. Meanwhile Anderson and Alagiyawadu [28] reported eleven mother–calf pairs of BWs in the same area during the month of April in a six-year period. Moreover, five instances of courtship behavior or male competitions were also observed (Per. Comm. Raja & the Whale), but within the year 2015, breeding signs were observed only during the end of December. These signs were characterized by the pairing of animals or two males with a female, males chased behind the female, the aggressive behavior of males, breeching and fast swimming, similar to the study of Schall et al. [55]. Previous studies, historical records and data collection for the current study (2015) postulate that sightings of mother–calf pairs most commonly occur between March and April during intermonsoonal seasons, which are potentially the most favorable calving months in Sri Lankan waters. The current study emphasizes that the south coastal region of Sri Lanka is utilized by the Indian Ocean BWs as a feeding, breeding and calving ground.

Previous scientific studies, contemporary line transect surveys and opportunistic sighting data revealed that the highest abundance estimates of BWs tended to overlap with the busy shipping lanes located between Dondra Head and Galle during the months from April to December. Increased vessel traffic travelling at high speeds significantly increases the likelihood of ship strikes in the near vicinity of dense aggregations of BWs foraging on prey items [4,14]. Over the last few decades, cetacean research efforts have steadily increased owing to the increased anthropogenic threats and a greater focus on the ecological and socioeconomic importance of BWs in Sri Lanka. Therefore, the design and implementation of conservation management policies are imperative for their protection and the prevention of anthropogenic threats to BW populations involving ship strikes, acoustic pollution and contaminants. Further boat-based surveys are vitally important in order to condense our knowledge gap with regards to the distributional patterns of BWs to identify critical foraging 'hotspots' and calving grounds and to delineate the potential seasonal movement patterns of BWs within Sri Lankan waters. Furthermore, digital marketing efforts, as effective communication, are of paramount importance on a global scale to promote knowledge of human–wildlife coexistence for conservation purposes [56].

## 5. Conclusions

The relatively recent identification of BW foraging and breeding 'hotspots' within the busy shipping lanes off Mirissa is currently experiencing a large expansion of commercial shipping, fishing activities and an increase in the domestic tourism sector, placing extensive pressure on the surrounding environment. Further legislative measures need to be implemented to reduce the risk of ship strikes to BW populations and to advance maritime safety for boat-based WW operations, as well as fishing vessels, scattered amongst the busy shipping traffic in the IOMMS. Moreover, the temporal and spatial variations in BW distribution and relative abundance could be positively used for the development of the whale-watching industry. Peak BW sighting periods with calm sea conditions (December to February) are more suitable for blue whale observations.

**Author Contributions:** Conceptualization, U.S.P.K.L. and P.K.P.B.T.; investigation, U.S.P.K.L.; writing original draft preparation, U.S.P.K.L.; writing—review and editing, U.S.A., K.A., M.H.R. and P.K.P.B.T.; supervision, U.S.A. and P.K.P.B.T. All authors have read and agreed to the published version of the manuscript.

**Funding:** This research received no external funding.

**Institutional Review Board Statement:** Not applicable.

**Informed Consent Statement:** Not applicable.

**Data Availability Statement:** Data available with the corresponding author.

**Acknowledgments:** The authors would like to thanks Rajesh Madushanka and other crew members of the Raja & the Whale, the commercial whale-watching company in Mirissa, Sri Lanka, and staff of the Kapparathota regional research center of the National Aquatic Resources Research and Development Agency for assisting data collection during the operations.

**Conflicts of Interest:** The authors declare that they have no conflict of interest.

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
