# Peer review of "Seasonal Occurrence of the Indian Ocean Blue Whale (Balaenoptera musculus indica) off South Coast of Sri Lanka"

_jmse, doi:10.3390/jmse11081523_

Round 1

Reviewer 1 Report

Dear Authors

I read your paper with great interest, and found it very interesting and relevant to readers and the scope of this journal. This is a very welcome study for the Sri Lankan region given the limited number of blue whale studies in that area. I found the title of the manuscript confusing since you talk much about CHL and SST throughout the manuscript, whales are there year-round despite changes in these variables, so how can you say that the presence of whales is due to these variables. I suggest that the title be revised- to describe the results of this. I don't have a lot of comments, other than the quality of Figures 2 and 3 needs to be improved so that they can be very visible. More comments are provided in the attached markup document.

The English language, but I have suggested few changes the attached markup document.

Author Response

Dear Reviewer

Authors are highly appreciated your constructive comments which supported the improved quality of MS

Reviewer 2 Report

This manuscripts needs an extensive English revision. Many sentences need to be rephrased since as of now, they lack sense. I recommend a careful revision of the manuscript, many words are not well written, verbs not correctly conjugated, whole sentences that have no link to the main text.

The methodology is not well explained. You do not mention any T-student test but then report it in the results. You only give p-values, but the readers need to see in a table a summary of your statistical analyses to understand what are you analysing.

Specific comments

Title: if you are citing a subspecies, name it in common language also, i.e. the Indian blue whale

Lines 42-44 no citation?

Lines 131-133 the crew is well experienced, but did you confirmed sightings?? With pictures, for example. It is you who is presenting the work

Line 348: incidents? 

Be consistent: ‘Raja and the Whale’ or ‘Raja & the Whale’

Results. You defended the experience of the crew, but then in Results state that some of the data were taken by other boats…and give no more information. Please clarify

Lines 385-388 conclusions…you are not testing these human impacts, how can you conclude saying they are increasing?

Figures 2 and 3 are totally blurred

Table 4 Remarks are not very informative, please clarify

The manuscript has many words not correctly written, and long sentences with erros that made them difficult to read, such as:

'Gill et al. [11] revealed that in Bonney upwelling region where blue whale largely used for feeding; sea surface temperature played a significant role among other various seascape variables that have been associated with upwelling'

'Furthermore, the adoption of unsustainable whale-watching practices due to extensive competition between boat operators during peak seasons is often encouraged resulting from economic pressures to achieve short-term benefits from 80 eco-tourism at the expense of displacing free-ranging BW into busy shipping lanes, and thus increasing the risk of ship collisions'

'In contrast, seven of thirteen records of mother-calf pairs (53%) were typically encountered in waters below 1000m and only one occurred in deeper location far away from the land. Moreover, breeching was frequently observed in the month of November to December. At the end of December group of around 20 members of BW were observed in five consecutive days showing signs of mating behaviors'

Author Response

Dear Reviewer,

We have highly appreciated your comments and also those are much more beneficial to improving the quality of MS.

Reviewer 3 Report

This manuscript (MS) aims to describes the distribution, abundance and seasonal variation of blue whales (BW) off the south coast of Sri Lanka in relation to potential anthropogenic threats from vessel traffic, acoustic pollution, and contaminants. Its primary contribution is the collation and presentation of whale-watching (WW) sighting data in the region. The MS is generally well written and well structured.

---

The introduction is largely comprehensive, although evidence of the increasing trend in vessel traffic (and the actual threat this poses to BW, e.g. ship strike data) is missing.

61-64: Some numeration/quantification/evidence of the volume of vessel traffic in the area is required in the body of the MS (not just references); particularly because the author's describe the volume as increasing. If this is the driving force behind the study and the recommendations made at the end, it is crucial to establish that vessel traffic is in fact increasing at the start of the MS.

---

The methods are sparse, but sufficient to understand the study. However, I have concerns about the choice of statistical analyses.

98:99: Requires reference.

102: Evidence of "increased risk of anthropogenic threats" required.

The data concerning the relationship between SST and Chlorophyll-a on BW monthly sightings were analysed using non-parametric Mann-Whitney tests. The devlopment of mixed models has made this/these type of tests largely obsolete and these data (as count data) would be much better analysed using some form of Poisson or negative binomial general linear mixed model. This would allow the authors to determine the relative importance (effect sizes) of each SST and Chlorophyll-a on BW abundance whilst taking into account season, month, and any other potential blocking variables (as random effects) such as crew identity or time of day.

Furthermore, by binning the continuous variables SST and Chlorophyll-a into 2 categories (high and low) the authors obscure a huge amount of potential variation and its impact on BW sightings. Using a mixed model would also allow the authors to include these predictors in their original, untransformed state. Binning SST and Chlorophyll-a is likely to elevate the chances of finding a significant effect where there may be none. If the authors do not wish to do this, at the very least I recommend they create low, middle, and high categories (1 standard deviation above and below the mean representing the boundaries perhaps) rather than simply high and low.

Also, can the authors be certain that the BW sightings recored by the WW crew were of different individuals each time? If not, they may be overestimating abundance and habitat selection by counting the same individuals repeatadly in a single day. Some mention or explanation of this is required please.

---

The results are generally clear, although the figures are of such low resolution that I cannot attest to their accuracy and/or usefulness. I realise this may not be the fault of the author's, but reviewer's need to be able to view all of the figures clearly in order to evaluate them and their contribution to the MS. These figures do look particularly interesting!

The data are summarised nicely in the tables.

Figure 4: Legend required for TSS.

243-244: Evidence required please.

---

The discussion is generally good. The authors do a good job of stating the limitations of the study. However, the point of each section in the discussion is not always clear. It would benefit from careful editing to ensure that every point made in the discussion leads/contributes to the broader aim of the MS.

Finally, the recommendations concerning vessel management etc. are sound in principle, but without a clear presentation of the volume of vessel traffic, shipping lanes, and how these interact with BW habitat selection and migration routes, I fear that the recommendations carry little weight.

Generally well written, but errors in English langugage appear throughout. The MS would benefit from further careful editing.

Author Response

Dear Reviewer

Authors are highly appreciated your comments since they were supported to improving the quality of MS
